# Examining Canadian youth's engagement with food companies via digital media

**Laura Vergeer[1], Meghan Pritchard[1], Carolina Soto[1], Elise Pauzé[2], Ashley Amson[2], Dana Lee Olstad[3], Monique Potvin Kent** [1]*

**1** School of Epidemiology and Public Health, Faculty of Medicine, University of Ottawa, Ottawa, Ontario, Canada, **2** Interdisciplinary School of Health Sciences, University of Ottawa, Ottawa, Ontario, Canada, **3** Department of Community Health Sciences, Cumming School of Medicine, University of Calgary, Calgary, Alberta, Canada

* monique.potvinkent@uottawa.ca

## Abstract

Digital food marketing to youth is concerning given its widespread reach, engagement strategies and influence on lifelong food behaviours. Nonethless, little is known about youth's engagement (i.e., liking/sharing/following food companies on social media, having food company/restaurant/delivery service apps downloaded) with food companies via digital media, particularly in Canada. This study examined whether youth's digital engagement with food companies differed by sociodemographic characteristics in Canada. An observational cross-sectional online survey was conducted in 2023 among 1162 Canadian children (aged 10–12 years) and adolescents (13–17 years). Participants self-reported their sociodemographic information and engagement with food companies via digital media. Descriptive analyses and logistic regression models examined differences in engagement by gender, age group, race/ethnicity and income adequacy. Among all participants, 20.9% reported having liked, shared, or followed food/restaurant companies on social media, 23.1% had food/restaurant company apps on their smartphones, and 16.6% had apps for food delivery services. White participants and youth from medium income adequacy households had lower odds of having liked/shared/followed food companies on social media than racial/ethnic minority group participants (OR: 0.59; 95% CI: 0.43, 0.80) and those from low income adequacy households (OR: 0.57; 95% CI: 0.41, 0.80), respectively. Children and White participants had lower odds of reporting food company apps on their smartphones than adolescents (OR: 0.54; 95% CI: 0.41, 0.72) and racial/ethnic minority group participants (OR: 0.48; 95% CI: 0.35, 0.64), respectively. Children and White participants also had lower odds of reporting food delivery service apps on their smartphones than adolescents (OR: 0.52; 95% CI: 0.38, 0.72) and racial/ethnic minority group participants (OR: 0.32; 95% CI: 0.23, 0.44), respectively. No significant differences were observed between genders. Overall, many Canadian youth are engaging with food companies via digital media. Government-led

**Data availability statement:** All relevant data are within the manuscript and its Supporting Information file.

**Funding:** This study was funded by a Canadian Institutes of Health Research (CIHR) Project Grant (PJT 178193; MPK). EP is funded by the Canada Graduate Scholarship to Honour Nelson Mandela awarded by CIHR (2019-22) and Fonds de recherche du Québec – santé (2022-24). The funders had no role in study design, data collection and analysis, decision to publish, or preparation of the manuscript.

**Competing interests:** I have read the journal's policy and the authors of this manuscript have the following competing interests: EP received has received an honorarium from the Stop Marketing to Kids Coalition (2018) and Heart & Stroke (2023) for doing policy and advocacy work related to food marketing to children. She was also recently (2023-24) employed by Heart & Stroke on a part-time basis. This work and compensation are not related to the current research. All remaining authors declare no conflicts of interest.

food marketing regulations that extend to social media and food company and delivery service apps are warranted.

---

## Author summary

Digital marketing of unhealthy foods to youth is a significant public health concern, yet few studies have examined how youth are engaging with food companies through digital media. Accordingly, we conducted a survey with 1162 Canadian youth aged 10–17 years to examine whether their engagement with food companies via digital media differed by age, gender, race/ethnicity or perceived income adequacy. Participants provided information about their sociodemographic characteristics and engagement with food companies via digital media (i.e., liking/sharing/following food companies on social media and having food company/restaurant/delivery service apps downloaded). Over one-fifth of the sample reported having liked, shared, or followed a food company on social media, and nearly one-quarter reported having food company apps on their smartphones. Fewer youth (17%) reported having food delivery service apps. Differences in digital engagement with food companies between sociodemographic groups were also observed. Government food marketing restrictions being considered in Canada should apply to digital media, including food company and food delivery service apps. These policies may include restrictions on the targeting and reach of food companies' social media posts and stricter age verification procedures on food company and food delivery service apps to help protect youth from unhealthy food marketing via digital media.

## Introduction

The consumption of unhealthy foods and beverages is widely recognized as a significant contributor to obesity and other non-communicable diseases (NCDs) [1]. Children and adolescents (hereafter referred to as "youth") are especially vulnerable to poor quality diets due to their frequent exposure to powerful marketing of unhealthy foods [2]. Youth's exposure to marketing of these products negatively influences their food choices, food purchasing behaviours (or purchase requests), and eating habits [3].

The rise of digital media and increasing amounts of time spent online among youth are concerning, given the superior reach of digital marketing compared to traditional marketing methods and evidence linking social media use with poor dietary habits [4,5]. Canadian youth in grades 6–12 spend 7–8 hours daily on digital devices, with 91% of Ontario students in grades 7–12 using social media daily and 31% spending 5 + hours on these platforms [6,7]. Unlike traditional forms of marketing, digital marketing allows for targeted advertising, making it more effective at engaging specific audiences [8]. Digital platforms are also different from other forms of media, as youth

can actively interact with brands on these sites by liking, sharing, or following their content [8]. This is concerning given the evidence to suggest that children who engage more food brands and content are more likely to consume unhealthy foods and beverages [9]. A Canadian study conducted in 2022 estimated that youth were exposed to 6,023 marketing instances per year [10]. Digital marketing in Canada is rapidly growing, with spending projected to rise from $16.2 billion in 2024 to over $21.5 billion by 2028 [11,12].

Studies have consistently demonstrated sociodemographic differences in youth's exposure to unhealthy food marketing, including via digital media. For example, Canadian studies found adolescents were more likely to be exposed to digital food marketing than children [13,14]. Gender also plays a role in youth's food marketing exposure. On social media, boys are more likely to be exposed to food marketing featuring male actors, male influencers, and appeals to achievement and athleticism, whereas girls are more likely to see food marketing containing quizzes, surveys or polls [15]. Additionally, boys more commonly follow food companies and share food posts [16]. There is also evidence of differences in youth's digital marketing exposure by race/ethnicity and perceived income adequacy. A recent study by our group found that youth of racial/ethnic minorities reported more frequent exposure to digital marketing of unhealthy foods than White youth in Canada [17]. Differences were also seen between income adequacy groups [17]. It is important to document sociodemographic differences in youth's food marketing exposure to understand targeted marketing and impacts on marketing perceptions, food purchases and consumption.

Engagement with food companies through digital media is becoming an increasingly common form of digital food marketing to youth [9,18]. Engaging with digital food and beverage marketing by liking, sharing, commenting or downloading apps can have negative impacts on behavioural and health outcomes, such as fostering brand loyalty and increased unhealthy food consumption [19]. Existing studies indicate many youths are engaging with food and beverage companies on social media platforms such as YouTube, Instagram, and TikTok, with negative impacts on their diet quality. A recent study in the United States found that 70% of adolescents reported liking, sharing, or following food and beverage brands on social media [18]. Another study found that higher levels of online engagement correlate with higher consumption of unhealthy foods and beverages [9]. These findings are concerning given that studies have found most advertised products on digital media are energy-dense, nutrient-poor foods [20,21]. Sociodemographic factors may also play a role in how youth engage with food and beverage marketing online. For example, a survey of American adolescents in 2017 found that non-Hispanic Black and less-acculturated Hispanic adolescents were more likely to like, share, or follow food brands on social media than non-Hispanic White adolescents [18]. The study also found that girls and adolescents whose parents had a high school diploma or less were more likely to like, share, or follow ≥5 food brands on social media, compared with boys and adolescents with college-educated parents, respectively. [18]. Another form of engagement with food companies via digital media is the use of food company (e.g., Red Bull TV app) or restaurant mobile apps (e.g. McDonald's app) and food delivery service apps (e.g., Uber Eats). Here, sociodemographic differences have been observed in app usage as well, although evidence is limited among youth. One study conducted a survey among young adults (aged 18–25 years) in the USA and found those who identified as non-Hispanic Black and Hispanic/Latinx reported using food delivery apps more frequently than White participants [22]. No differences were observed between sexes, but having higher perceived subjective social status was associated with more frequent use of food delivery apps.

It is crucial to examine youth's engagement with digital media because digital platforms are a source of marketing and often collect information from consumers to enable highly targeted and personalized marketing, often through artificial intelligence [23]. Considering the widespread and frequent use of digital media among youth, it is also important to understand how different sociodemographic factors influence online engagement with food and beverage companies. Limited research has explored whether sociodemographic characteristics play a role in how youth engage with food companies via digital media, and there is no existing research on this topic in Canada. This study aimed to address this gap by examining whether youth engagement with food and beverage companies via digital media differs by sociodemographic characteristics in Canada. For this study, "engagement" encompassed both liking, sharing, or following food, beverage, or

restaurant companies on social media, as well as having the apps of food or restaurant companies and/or food delivery services downloaded to the participant's smartphone.

## Materials and methods

### Ethics statement

This study was reviewed by and received ethics clearance through the University of Ottawa Research Ethics Board (file H-11-21-7343). Informed parental consent and youth assent were obtained for all participants prior to completing the survey.

### Study design

This study utilized data from a cross-sectional observational survey conducted among youth 10–17 years-old residing in the Canadian provinces of Ontario and Quebec, and followed the STROBE reporting guideline. Data were collected in April 2023 through a self-administered online questionnaire. Survey participants were recruited by Leger Marketing [24] and its affiliated partner Quest Mindshare [25]. Recruitment was stratified by sex (male, female), age group (10–12 years, 13–17 years) and province (Ontario, Quebec) [27]. Parents/guardians on the survey panel received email invitations to enroll their child in the study. To be eligible, youth participants had to reside in Ontario or Quebec, be between 10 and 17 years of age, and have access to a digital device to complete the online survey. Only one child per household was eligible to participate. Parents/guardians provided consent for their child's participation and completed a section of the survey providing household sociodemographic information, including income adequacy and the child's race/ethnicity. Youth subsequently completed the survey independently online.

The survey was developed using items from prior research, including the International Food Policy Study Youth Survey [26], the Canadian census, as well as other digital marketing studies [9,18] and prior food marketing research conducted by our group with Canadian youth [14]. A copy of the full survey is published elsewhere [27,28]. Quality control measures included filtering questions to flag non-sensical or gibberish responses and comparisons of responses to similar closed- and open-ended questions to ensure coherence. Participants who completed the survey in less than one-third of the median completion time were excluded. All survey measures included "prefer not to answer" as a response option. Respondents received either cash or virtual incentives redeemable for gift cards upon survey completion.

### Measures

**Sociodemographic characteristics.** Parents reported their child's sex at birth (male or female). Gender was assessed by asking youth participants, "What gender do you identify as?". Response options included "Boy," "Girl," and "I identify as …" (open-ended). Due to the small number of responses in the "Other" category, individuals who identified as "Trans" (n = 1) were assigned to the gender category opposite to their sex at birth and those who identified as "Non-binary" or with other minority gender identities (n = 4) were equally distributed into the male and female categories through random assignment following an approach proposed by Statistics Canada [29,30]. Age group was categorized into two groups: 10–12 years and 13–17 years. Province of residence was recorded as Ontario or Quebec. Parents/guardians identified their child's race/ethnicity from a list of categories as follows: Black (e.g., African, Afro-Caribbean, African Canadian); East Asian (e.g., Chinese, Korean, Japanese, Taiwanese descent); South Asian (e.g., Indian, Pakistani, Bangladeshi, Sri Lankan, Indo-Caribbean); Southeast Asian (e.g., Cambodian, Filipino, Indonesian, Thai, Vietnamese, or other Southeast Asian descent); Indigenous (e.g., First Nations, Métis, Inuk/Inuit); Latin American (e.g., Latin American, Hispanic); Middle Eastern (e.g., Arab, Persian, West Asian descent including Afghan, Egyptian, Iranian, Lebanese, Turkish, Kurdish); White (e.g., European descent); or another race category (please specify). Due to small

numbers of observations in the racial/ethnic minority categories, responses were collapsed into two categories: "White" and "racial/ethnic minority". Income adequacy was assessed by asking parents how difficult it is to make ends meet based on their monthly income ("very difficult", "difficult", "neither easy nor difficult", "easy" or "very easy"). Responses were collapsed into three categories: low ("very difficult" and "difficult"), medium ("neither easy nor difficult"), and high income adequacy ("easy" and "very easy").

**Engagement with food, beverage, and restaurant companies.** Participants were asked: "Have you ever liked, shared, or followed any food or beverage companies (e.g., McDonald's, Coca-Cola, or restaurants) on social media?" (response options included "yes" and "no"). Respondents who selected "yes" were further asked to specify which food and beverage (hereafter referred to as "food") or restaurant companies they had liked, shared, or followed and their reasons for following companies on social media. Reasons for following companies included: "I get deals on products"; "I like to find out about new products"; "The posts are funny"; "The posts are interesting"; "The posts that show food and beverages look really good, tasty, or delicious"; "The posts tell me about special events in my city or town"; "I can win contests or prizes"; "I can donate to worthy causes"; "I like the posts"; and "I like to share these posts with my friends".

Additionally, participants were asked whether they had any restaurant or food company apps (e.g., McDonald's or RedBull TV app) on their smartphones, excluding food delivery service apps. Those who answered "yes" were asked to list the specific apps. Similarly, participants were also asked whether they had any food delivery service apps on their smartphones. Respondents who answered "yes" were asked to select all apps they had on their smartphones. Response options included: "Uber Eats"; "Skip the Dishes"; "Grubhub"; "DoorDash"; "Foodora"; or "Other. Please specify".

## Statistical analysis

The approach used to derive the analytic study sample(s) is shown in Fig 1, with sub-samples used for some descriptive analyses (described in detail below). In total, 1211 participants completed the survey. Participants who selected "prefer not to answer" in response to any of the questions about sociodemographic characteristics, liking, sharing or following food companies on social media, and whether they had food company apps or food delivery service apps on their smartphones were excluded (n = 49), resulting in an analytic sample size of 1162 participants.

Descriptive statistics (e.g., frequencies, percentages) were calculated for outcome variables and presented by sociodemographic characteristics (e.g., gender, age group, race/ethnicity and income adequacy). This included the number and percentage of participants who reported having ever liked, shared, or followed any food or restaurant companies on social media, who reported having apps for restaurants or food companies on their smartphone, and who had any food delivery service apps on their smartphone. The number of food and restaurant companies liked, shared or followed on social media, the companies most liked, shared or followed, and reasons for following food or restaurant companies on social media was also examined among youth who reported having ever liked, shared or followed food or restaurant companies on social media (n = 243). Additionally, the number of food and restaurant company apps downloaded and the most common food and restaurant company apps on the smartphones of participants who reported having food or restaurant company apps on their smartphones (n = 268) were calculated. The number and proportion of participants who reported having apps for food delivery services on their smartphone (n = 193), including Uber Eats, Skip the Dishes, Grubhub, DoorDash, Foodora or another food delivery service app, were also determined.

Binary logistic regression models were used to examine differences in reporting: i) having liked, shared, or followed food or restaurant companies on social media; ii) having food or restaurant company app(s) on their smartphone; and iii) having food delivery service app(s) on their smartphone among sociodemographic subgroups (n = 1162). Predictors examined included gender (boy, girl), age group (10–12 years, 13–17 years), race/ethnicity (White, racial/ethnic minority) and income adequacy (low, medium, high). Results were considered statistically significant when p < 0.05. All assumptions of logistic regression were met (multicollinearity was assessed using Variance Inflation Factors). All analyses were completed using IBM SPSS Statistics (Version 29.0.1.0).

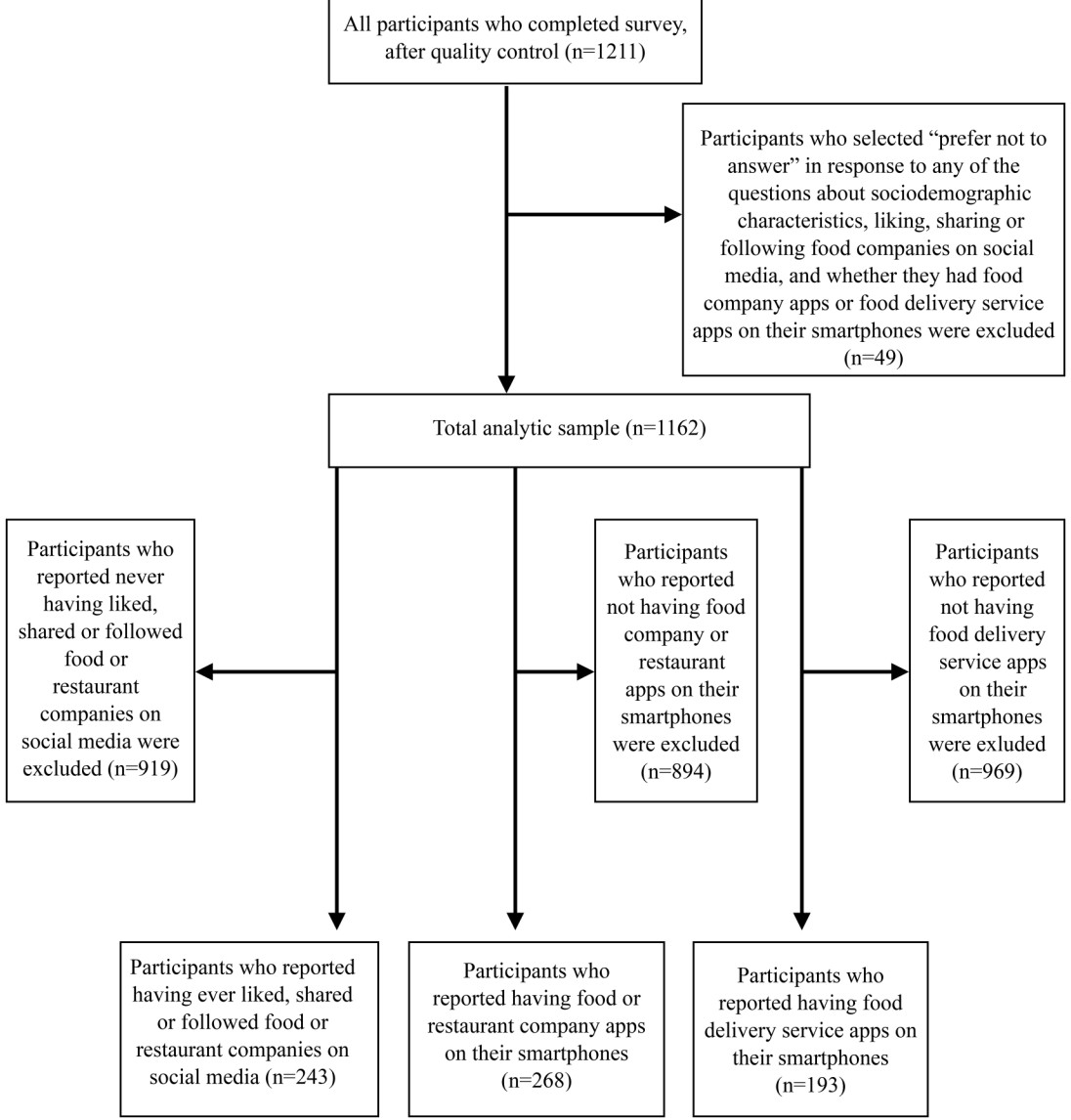

**Fig 1. The approach used to derive the analytic study sample(s).** Sub-samples were examined for the descriptive analyses presented in Tables 3, 4, 5.

## Results

### Sociodemographic characteristics

Among the total sample (n = 1162), there was an approximately even split by gender (51.0% identified as boys) and age group (50.0% were 13–17 years-old). Three-quarters of the sample (75.5%) identified as White. Additionally, 38.7% of participants were from households reporting medium income adequacy; 33.7% and 27.5% were from households of low and high income adequacy, respectively.

## Liking, sharing, or following food and restaurant companies on social media

Overall, 20.9% of participants reported ever having liked, shared, or followed food or restaurant companies on social media (Table 1). Similar results were observed between genders (21.1% of boys and 20.7% of girls) and age groups (18.8% of children and 23.0% of adolescents). By race/ethnicity, 28.1% of racial/ethnic minority group participants reported having liked, shared, or followed food or restaurant companies on social media, compared with only 18.6% of White participants. Approximately one-quarter (25.5%) of participants with low income adequacy reported having liked, shared, or followed food or restaurant companies on social media, compared with 21.6% of participants with high income adequacy, and 16.4% of those with medium income adequacy.

The logistic regression model found that White participants had lower odds of reporting having liked, shared, or followed food or restaurant companies on social media than racial/ethnic minority group participants (AOR: 0.59; 95% CI: 0.43, 0.80; Table 2). Moreover, youth from households of medium income adequacy had lower odds than those from low income adequacy households to report having liked, shared, or followed food or restaurant companies on social media (AOR: 0.57; 95% CI: 0.41, 0.80). Differences between genders (p = 0.82) or age groups (p = 0.10) were not statistically significant.

**Table 1. The number and proportion of youth in Canada who reported having liked, shared or followed food or restaurant companies on social media, having apps for food or restaurant companies on their smartphones, and/or having apps for food delivery services on their smartphones, presented by sociodemographic characteristics (n=1162).**

|  | Gender | | Age group (years) | | Race | | Perceived income adequacy | | | Total (n=1162) |
|---|---|---|---|---|---|---|---|---|---|---|
|  | Boys (n=593) | Girls (n=569) | 10-12 (n=580) | 13-17 (n=582) | White (n=877) | Minority (n=285) | Low (n=392) | Medium (n=450) | High (n=320) |  |
|  | % (n) | % (n) | % (n) | % (n) | % (n) | % (n) | % (n) | % (n) | % (n) | % (n) |
| Liked, shared or followed food/restaurant companies on social media | 21.1 (125) | 20.7 (118) | 18.8 (109) | 23.0 (134) | 18.6 (163) | 28.1 (80) | 25.5 (100) | 16.4 (74) | 21.6 (69) | 20.9 (243) |
| Apps for food/restaurant companies on smartphone | 22.4 (133) | 23.7 (135) | 17.6 (102) | 28.5 (166) | 19.5 (171) | 34.0 (97) | 26.3 (103) | 19.8 (89) | 23.8 (76) | 23.1 (268) |
| Apps for food delivery services on smartphone | 15.2 (90) | 18.1 (103) | 12.1 (70) | 21.1 (123) | 12.1 (106) | 30.5 (87) | 18.9 (74) | 14.9 (67) | 16.3 (52) | 16.6 (193) |

**Table 2. The odds of reporting having liked, shared or followed food companies on social media, having restaurant or food company app(s) on their smartphone, and having food delivery service app(s) on their smartphone among youth in Canada (n=1162).**

|  | Liked, shared or followed food companies on social media | | Has restaurant or food company app(s) on smartphone | | Has food delivery service app(s) on smartphone | |
|---|---|---|---|---|---|---|
|  | χ2, p-value[a] | Adjusted OR (95% CI)[a,b] | χ2, p-value[a] | Adjusted OR (95% CI)[a,b] | χ2, p-value[a] | Adjusted OR (95% CI)[a,b] |
| Gender | 0.054, p=0.82 |  | 0.16, p=0.69 |  | 1.17, p=0.28 |  |
| Boy vs. girl |  | 1.04 (0.78, 1.38) |  | 0.95 (0.72, 1.25) |  | 0.84 (0.61, 1.15) |
| Age group (years) | 2.68, p=0.10 |  | **17.93, p<0.001** |  | **15.36, p<0.001** |  |
| 10-12 vs. 13-17 |  | 0.79 (0.59, 1.05) |  | **0.54 (0.41, 0.72)** |  | **0.52 (0.38, 0.72)** |
| Race/ethnicity | **11.35, p<0.001** |  | **23.38, p<0.001** |  | **46.63, p<0.001** |  |
| White vs. minority |  | **0.59 (0.43, 0.80)** |  | **0.48 (0.35, 0.64)** |  | **0.32 (0.23, 0.44)** |
| Income adequacy | **10.67, p=0.005** |  | 5.51, p=0.06 |  | 2.89, p=0.24 |  |
| High vs. low |  | 0.80 (0.56, 1.14) |  | 0.86 (0.60, 1.21) |  | 0.82 (0.55, 1.22) |
| Medium vs. low |  | **0.57 (0.41, 0.80)** |  | 0.68 (0.49, 0.94) |  | 0.73 (0.50, 1.06) |

[a]Binary logistic regression models included gender, age group, race/ethnicity and income adequacy. Boldface indicates statistical significance at an alpha level of 0.05. [b]The reference category is listed second.

Of the participants who reported having liked, shared, or followed food or restaurant companies on social media (n = 243), most reported having liked, shared, or followed at least one food company (59.3%), with fewer participants having reportedly liked, shared, or followed at least two (9.1%), three (7.0%) or four (1.2%) companies (Table 3). Overall, there were relatively small differences by sociodemographic characteristics; however, some notable differences were observed. For example, 65.0% of participants from low income adequacy households reported having liked, shared, or followed at least one food company on social media, compared with only 50.7% of participants from high income adequacy households.

The three most popular companies among participants who reported having liked, shared, or followed food or restaurant companies on social media were McDonald's (57.2%), Coca-Cola (8.6%), and Tim Hortons (6.6%; Table 3). In terms of gender, similar results were observed for McDonald's and Coca-Cola, but 10.2% of girls reported having liked, shared, or followed Tim Hortons on social media, compared with only 3.2% of boys. Between age groups, 60.1% of adolescents reported having liked, shared, or followed McDonald's on social media, compared with only 41.1% of children. Similar results were observed for race/ethnicity, with 60.1% of White participants reporting having liked, shared, or followed McDonald's on social media, compared with 51.3% of racial/ethnic minority group participants. Differences in liking, sharing, or following McDonald's were also seen by income adequacy group, with 62.2% of participants from medium income adequacy households reporting having liked, shared, or followed this company on social media, compared with 52.2% of participants from high income adequacy households.

Among participants who reported following food or restaurant companies on social media (n = 243), the most commonly reported reasons for following them were "the posts that show food and beverages look really good/tasty/delicious" (37.9%), and "I like to find out about new products" (30.5%), and "I get deals on products" (28.4%; Table 3). Nearly half (48.3%) of girls reported following food companies on social media because the foods look good, compared with only 28.0% of boys. Similarly, 36.3% of racial/ethnic minority group participants followed food companies on social media to find out about new products, compared with 27.6% of White participants. In addition, 32.8% of adolescents reported following food companies on social media to get deals on products, compared with only 22.9% of children.

### Apps for food/restaurant companies on smartphone

Within the total sample, 23.1% of participants reported having apps for food or restaurant companies on their smartphone (Table 3). Notably, 34.0% of youth identifying as a racial/ethnic minority group reported having food company apps on their smartphone, compared with only 19.5% of participants identifying as White. Moreover, 28.5% of adolescents reported having food company apps, compared with only 17.6% of children. In terms of income adequacy, participants from low income adequacy households most commonly had food company apps on their smartphones (26.3%), followed by those from high (23.8%) and medium income adequacy households (19.8%). The prevalence was similar between genders (22.4% of boys and 23.7% of girls).

The logistic regression model found that children had lower odds of having food company apps on their smartphone than adolescents (OR: 0.54; 95% CI: 0.41, 0.72; Table 2). Additionally, White participants had lower odds of reporting food company apps on their smartphone than racial/ethnic minority group participants (OR: 0.48; 95% CI: 0.35, 0.64). No statistically significant differences were observed between genders (p = 0.69) or income adequacy groups (p = 0.06).

Among participants who reported having food company apps on their smartphone (n = 268), 52.6% had at least one food company app, while 39.2% had at least two (Table 3). When examined by sociodemographic characteristics, the greatest differences were observed between age groups and income adequacy groups. For example, 43.4% of adolescents reporting having at least two food company apps on their smartphone, compared with only 32.4% of children. In addition, 60.2% of participants from low income adequacy households reported having at least one food company app on their smartphone, compared with only 42.1% of participants from households with high income adequacy.

**Table 3. The number of food and restaurant companies liked, shared or followed, companies most liked/shared/followed, and reasons for following companies on social media among youth who reported having ever liked, shared or followed any of these companies on social media (n = 243)[a].**

| | Gender | | Age group (years) | | Race/ethnicity | | Perceived income adequacy | | | Total (n=243) |
|---|---|---|---|---|---|---|---|---|---|---|
| | Boys (n=125) | Girls (n=118) | 10-12 (n=109) | 13-17 (n=134) | White (n=163) | Minority (n=80) | Low (n=100) | Medium (n=74) | High (n=69) | |
| | % (n) | % (n) | % (n) | % (n) | % (n) | % (n) | % (n) | % (n) | % (n) | % (n) |
| Number of companies liked, shared or followed on social media[b] | | | | | | | | | | |
| At least one | 60.0 (75) | 58.5 (69) | 62.4 (68) | 56.7 (76) | 62.0 (101) | 53.8 (43) | 65.0 (65) | 59.5 (44) | 50.7 (35) | 59.3 (144) |
| At least two | 8.0 (10) | 10.2 (12) | 6.4 (7) | 11.2 (15) | 9.2 (15) | 8.8 (7) | 6.0 (6) | 12.2 (9) | 10.1 (7) | 9.1 (22) |
| At least three | 8.0 (10) | 5.9 (7) | 6.4 (7) | 7.5 (10) | 6.1 (10) | 8.8 (7) | 7.0 (7) | 8.1 (6) | 5.8 (4) | 7.0 (17) |
| At least four | 0.8 (1) | 1.7 (2) | 0.9 (1) | 1.5 (2) | 1.8 (3) | 0.0 (0) | 0.0 (0) | 1.4 (1) | 2.9 (2) | 1.2 (3) |
| Unspecified[c] | 23.2 (29) | 23.7 (28) | 23.9 (26) | 23.1 (31) | 20.9 (34) | 28.7 (23) | 22.0 (22) | 18.9 (14) | 30.4 (21) | 23.5 (17) |
| Companies most liked, shared or followed on social media | | | | | | | | | | |
| McDonald's | 58.4 (73) | 55.9 (66) | 61.5 (67) | 41.1 (72) | 60.1 (98) | 51.3 (41) | 57.0 (57) | 62.2 (46) | 52.2 (36) | 57.2 (139) |
| Coca-Cola | 9.6 (12) | 7.6 (9) | 7.3 (8) | 7.4 (13) | 8.6 (14) | 8.8 (7) | 10.0 (10) | 8.1 (6) | 7.2 (5) | 8.6 (21) |
| Tim Hortons | 3.2 (4) | 10.2 (12) | 3.5 (5) | 6.3 (11) | 6.1 (10) | 7.5 (6) | 5.0 (5) | 6.8 (5) | 8.7 (6) | 6.6 (16) |
| Other[d] | 55.2 (69) | 52.5 (62) | 47.7 (52) | 59.0 (79) | 51.5 (84) | 58.8 (47) | 48.0 (48) | 54.1 (40) | 62.3 (43) | 53.9 (131) |
| Reasons for following companies on social media | | | | | | | | | | |
| I get deals on products | 27.2 (34) | 29.7 (35) | 22.9 (25) | 32.8 (44) | 31.3 (51) | 22.5 (18) | 33.0 (33) | 18.9 (14) | 31.9 (22) | 28.4 (69) |
| I like to find out about new products | 28.0 (35) | 33.1 (39) | 25.7 (28) | 34.3 (46) | 27.6 (45) | 36.3 (29) | 29.0 (29) | 33.8 (25) | 29.0 (20) | 30.5 (74) |
| The posts are funny | 23.2 (29) | 23.7 (28) | 25.7 (28) | 21.6 (29) | 22.1 (36) | 26.3 (21) | 23.0 (23) | 23.0 (17) | 24.6 (17) | 23.5 (57) |
| The posts are interesting | 24.8 (31) | 28.8 (34) | 30.3 (33) | 23.9 (32) | 24.5 (40) | 31.3 (25) | 27.0 (27) | 23.0 (17) | 30.4 (21) | 26.7 (65) |
| The posts that show food and beverages look really good/tasty/delicious | 28.0 (35) | 48.3 (57) | 40.4 (44) | 35.8 (48) | 40.5 (66) | 32.5 (26) | 39.0 (39) | 37.8 (28) | 36.2 (25) | 37.9 (92) |
| The posts tell me about special events in my city or town | 11.2 (14) | 11.9 (14) | 11.9 (13) | 11.2 (15) | 8.6 (14) | 17.5 (14) | 16.0 (16) | 6.8 (5) | 10.1 (7) | 11.5 (28) |
| I can win contests or prizes | 23.2 (29) | 29.7 (35) | 22.9 (25) | 29.1 (39) | 26.4 (43) | 26.3 (21) | 32.0 (32) | 21.6 (16) | 23.2 (16) | 26.3 (64) |
| I can donate to worthy causes | 4.8 (6) | 5.9 (7) | 7.3 (8) | 3.7 (5) | 4.9 (8) | 6.3 (5) | 5.0 (5) | 1.4 (1) | 10.1 (7) | 5.3 (13) |
| I like the posts | 27.2 (34) | 22.0 (26) | 28.4 (31) | 21.6 (29) | 25.8 (42) | 22.5 (18) | 24.0 (24) | 24.3 (18) | 26.1 (18) | 24.7 (60) |
| I like to share these posts with my friends | 14.4 (18) | 25.4 (30) | 19.3 (21) | 20.1 (27) | 16.0 (26) | 27.5 (22) | 17.0 (17) | 18.9 (14) | 24.6 (17) | 19.8 (48) |
| Other | 4.0 (5) | 2.5 (3) | 2.8 (3) | 3.7 (5) | 3.1 (5) | 3.8 (3) | 6.0 (6) | 0.0 (0) | 2.9 (2) | 3.3 (8) |

[a]Only participants who indicated that they had ever liked, shared or followed any food or beverage companies on social media (n = 243) were asked which companies they had liked, shared or followed and why. [b]The number of companies are reported as "at least" because some open-ended responses included a mixture of identifiable food company names and unidentifiable responses (e.g., "McDonald's" followed by a series of symbols). [c]Refers to participants who indicated they had liked, shared or followed any food or beverage companies on social media but did not specify which companies they had liked, shared or followed.

Among participants who reported having at least one food company app on their smartphones (n = 268), the apps most reported were for McDonald's (69.8%), Tim Hortons (32.1%), and Starbucks (7.8%; Table 3). Overall, there were relatively small differences in the proportion of youth with apps for these companies on their smartphones by gender, race/ethnicity, or income adequacy group. Greater differences were observed between age groups. For example, 36.7% of adolescents reported having the Tim Hortons app on their smartphones, compared with only 24.5% of children.

**Food delivery service apps on smartphone**

Among the total sample, 16.6% of youth reported having food delivery service apps on their smartphone (Table 1). Similar findings were observed between genders, with 15.2% of boys and 18.1% of girls reportedly having food delivery service apps on their smartphones. However, food delivery apps on smartphones were more commonly reported by adolescents (21.1%) than children (12.1%). In addition, 30.5% of youth identifying as a racial/ethnic minority group reported having food delivery service apps on their smartphones, compared with only 12.1% of White participants. Similar results were observed between youth from households with low (18.9%), medium (14.9%) and high (16.3%) income adequacy in terms of whether they reported having food delivery service apps on their smartphones.

The logistic regression model found that children had lower odds of reporting having food delivery service apps on their smartphone than adolescents (OR: 0.52; 95% CI: 0.38, 0.72; Table 2). White participants also had lower odds of reporting food delivery service apps than racial/ethnic minority group participants (OR: 0.32; 95% CI: 0.23, 0.44). No statistically significant differences were observed between genders (p = 0.28) or income adequacy groups (p = 0.24).

Of participants who reported having food delivery service apps on their smartphone (n = 193, 16.6% of total sample), 71.5% reported having the Uber Eats app downloaded (Table 4). Skip the Dishes (46.1%), and DoorDash (48.7%) were also popular, while fewer participants reported having Foodora (3.1%), Grubhub (2.6%), or other food delivery service apps (2.1%) on their smartphone. Differences were observed across various sociodemographic groups for specific apps. For example, 80.0% of boys reported having the Uber Eats app on their smartphone, compared with 64.1% of girls. Similarly, the DoorDash app was reported to be downloaded to the smartphones of 58.6% of children, but only 43.1% of adolescents. It was also found that 82.8% of youth identifying as a racial/ethnic minority group had the Uber Eats app on their smartphone, compared with 62.3% of White participants. Lastly, the DoorDash app was reported to be on the smartphones of 58.1% of participants from households with low income adequacy, compared with 46.3% and 38.5% of youth from households with medium and high income adequacy, respectively.

**Table 4. The number of food and restaurant company apps downloaded and the most common food and restaurant company apps on the smartphones of youth who reported having apps for one or more food or restaurant companies on their smartphones (n = 268)[a].**

| | Gender | | Age group (years) | | Race/ethnicity | | Perceived income adequacy | | | Total (n=268) |
|---|---|---|---|---|---|---|---|---|---|---|
| | Boys (n=133) | Girls (n=135) | 10-12 (n=102) | 13-17 (n=166) | White (n=171) | Minority (n=97) | Low (n=103) | Medium (n=89) | High (n=76) | |
| | % (n) | % (n) | % (n) | % (n) | % (n) | % (n) | % (n) | % (n) | % (n) | % (n) |
| Number of food company apps on smartphone[b] | | | | | | | | | | |
| At least one | 53.4 (71) | 51.9 (70) | 58.8 (60) | 48.8 (81) | 53.8 (92) | 50.5 (49) | 60.2 (62) | 52.8 (47) | 42.1 (32) | 52.6 (141) |
| At least two | 41.4 (55) | 37.0 (50) | 32.4 (33) | 43.4 (72) | 40.4 (69) | 37.1 (36) | 34.0 (35) | 40.4 (36) | 44.7 (34) | 39.2 (105) |
| Unspecified[c] | 5.3 (7) | 11.1 (15) | 8.8 (9) | 7.8 (13) | 5.8 (10) | 12.4 (12) | 5.8 (6) | 6.7 (6) | 13.2 (10) | 8.2 (22) |
| Most common food company apps on smartphone | | | | | | | | | | |
| McDonald's | 72.2 (96) | 67.4 (91) | 75.5 (77) | 66.3 (110) | 71.3 (122) | 67.0 (65) | 71.8 (74) | 69.7 (62) | 67.1 (51) | 69.8 (187) |
| Tim Hortons | 33.1 (44) | 31.1 (42) | 24.5 (25) | 36.7 (61) | 33.9 (58) | 28.9 (28) | 28.2 (29) | 33.7 (30) | 35.5 (27) | 32.1 (86) |
| Starbucks | 4.5 (6) | 11.1 (15) | 2.0 (2) | 11.4 (19) | 7.6 (13) | 8.2 (8) | 8.7 (9) | 7.9 (7) | 6.6 (5) | 7.8 (21) |
| Other | 31.6 (42) | 26.7 (36) | 29.4 (30) | 28.9 (48) | 27.5 (47) | 32.0 (31) | 24.3 (25) | 29.2 (26) | 35.5 (27) | 29.1 (78) |

[a]Only participants who reported they had apps for restaurant or food/beverage companies on their smartphone (n = 268) were asked to report which companies' apps they had on their smartphone. [b]The number of companies are reported as "at least" because some open-ended responses included a mixture of identifiable food company names and unidentifiable responses (e.g., "McDonald's" followed by a series of symbols). [c]Refers to participants who indicated they had apps for restaurant or food/beverage companies on their smartphone but did not specify which companies' apps they had downloaded.

## Discussion

This study provided a novel examination of youth's engagement with food companies via digital media (i.e., liking/sharing/following food companies on social media and having food company/restaurant/delivery apps downloaded). More than one-fifth of the sample reported having liked, shared, or followed a food company on social media, and nearly one-quarter reportedly had food company apps on their smartphones. Fewer youth (17%) reported having food delivery service apps on their smartphones. Differences in engagement with food company apps between sociodemographic groups were also identified.

### Engagement through liking, sharing, following

Of the total sample, 21% reported having liked, shared, or followed food or restaurant companies on social media. Food companies' social media accounts commonly use photos, videos, links, and hashtags to promote their brands/products, often featuring promotional offers, gifts or competitions, famous people, branded or licensed characters, and/or brand elements (e.g., logos, colours, slogans), among other strategies [31]. Previous research has found that children who engaged more with food brands online in the form of reading or commenting on posts and 'liking' or 'hash tagging' brands are more likely to consume unhealthy food and beverages (after adjusting for age, sex and socioeconomic status) [9].

In the present study, liking, sharing, or following food or restaurant companies on social media was particularly common among youth identifying as a racial/ethnic minority group and those from households with low income adequacy. This finding aligns with a previous study of American adolescents (aged 13–17 years) that found non-Hispanic Black and less-acculturated Hispanic adolescents were more likely than non-Hispanic White adolescents to like, share, or follow food/beverage brands on social media [18]. The study also found no differences in liking, sharing, or following food brands on social media between younger (aged 13–14 years) and older adolescents (aged 15–17 years). This aligns with our findings and suggests age is not predictive of whether youth are likely to like, share, or follow food companies on social media. Similarly, no difference was observed between genders, indicating that boys and girls have similar odds of liking, sharing, or following food companies on social media. We did, however, observe differences between genders and age groups in the companies liked, shared, or followed and youth's reasons for following food companies on social media. For example, 60% of adolescents reporting having liked, shared, or followed McDonald's on social media, compared with only 41% of children, while 48% of girls and 28% of boys reported following food companies on social media because the food looks good. Our findings suggest a need for research to further explore differences in these behavioural nuances between sociodemographic groups, such as through qualitative studies. Understanding the food companies that youth interact with

**Table 5. The number and proportion of youth in Canada who reported having apps for food delivery services on their smartphone who indicated having one or more of the apps listed below, presented by sociodemographic characteristics (n = 193).**

| App | Gender | | Age group (years) | | Race/ethnicity | | Perceived income adequacy | | | Total (n=193) |
|---|---|---|---|---|---|---|---|---|---|---|
| | Boys (n=90) | Girls (n=103) | 10-12 (n=70) | 13-17 (n=123) | White (n=106) | Minority (n=87) | Low (n=74) | Medium (n=67) | High (n=52) | |
| | % (n) | % (n) | % (n) | % (n) | % (n) | % (n) | % (n) | % (n) | % (n) | % (n) |
| Uber Eats | 80.0 (72) | 64.1 (66) | 65.7 (46) | 74.8 (92) | 62.3 (66) | 82.8 (72) | 68.9 (51) | 74.6 (50) | 71.2 (37) | 71.5 (138) |
| Skip the Dishes | 43.3 (39) | 48.5 (50) | 52.9 (37) | 42.3 (52) | 46.2 (49) | 46.0 (40) | 50.0 (37) | 47.8 (32) | 38.5 (20) | 46.1 (89) |
| Grubhub | 4.4 (4) | 1.0 (1) | 4.3 (3) | 1.6 (2) | 1.9 (2) | 3.4 (3) | 2.7 (2) | 3.0 (2) | 1.9 (1) | 2.6 (5) |
| DoorDash | 51.1 (46) | 46.6 (48) | 58.6 (41) | 43.1 (53) | 50.0 (53) | 47.1 (41) | 58.1 (43) | 46.3 (31) | 38.5 (20) | 48.7 (94) |
| Foodora | 2.2 (2) | 3.9 (4) | 4.3 (3) | 2.4 (3) | 0.9 (1) | 5.7 (5) | 4.1 (3) | 0.0 (0) | 5.8 (3) | 3.1 (6) |
| Other | 2.2 (2) | 1.9 (2) | 2.9 (2) | 1.6 (2) | 0.9 (1) | 3.4 (3) | 1.4 (1) | 1.5 (1) | 3.8 (2) | 2.1 (4) |

aOnly participants who reported they had food delivery service apps on their smartphone (n = 193) were asked to indicate which apps they had.

on social media and their reasons for doing so provides insight into existing disparities in youth food marketing exposure, and may inform policies and regulations aimed at limiting youth's exposure to food marketing in digital media.

To our knowledge, no previous studies have examined differences in interaction with food companies via digital media among youth of households with varying income levels. Our finding that youth from medium income adequacy households had lower odds of liking, sharing, or following food companies on social media than those of low income households may be partly explained by the association between lower income and greater screen time that has been previously documented [32]. Additional research is needed to examine possible sociocultural or other systemic factors that may explain differences in digital engagement with food companies among youth of different sociodemographic groups.

## Engagement through having food company and food delivery service apps on smartphone

This study found that 23.1% of the sampled youth had apps for food companies on their smartphone, while 17% reported having food delivery service apps. The odds of having food company and food delivery apps on their smartphone were higher among adolescents than children, and among youth identifying as a racial/ethnic minority group, compared with those identifying as White. This difference between age groups may be related to the fact that food company apps are typically not intended for children under 13 years of age. A recent content analysis of the privacy policies and terms of service agreements for the mobile apps of 26 top Canadian fast food and dine-in restaurants found that 46% of food companies indicated the age of intended users for their apps, and 39% of these specified that their app was not meant for children younger than 13 years [33]. Nonetheless, none of the food companies examined in that study had a mandatory age verification process, enabling children to download and use food company apps irrespective of their age. Another explanation for our finding may be that adolescents are more likely than children to have disposable income, providing a means of using food company or delivery service apps independently without parental involvement [34]. In the USA, food accounts for approximately 20% of adolescents' total expenditures [35], much of which likely occurs via food delivery service or restaurant apps.

Our finding of differences between race/ethnicity groups is consistent with the higher levels of liking, sharing, or following food companies on social media among youth from racial/ethnic minority groups observed in previous research [32]. This may be related to racialized targeting of food marketing or app downloads, or the fact that most racial/ethnic minority individuals in Canada reside in large urban centres, which are also the most densely populated with restaurants and food delivery services [36,37]. Our results are also supposed by previous research indicating that young adults (aged 18–25 years) identifying as non-Hispanic Black and Hispanic/Latinx used food delivery services more frequently than participants who identified as White [22]. While this study examined a younger age group (youth aged 10–17 years) and did not examine frequency of food delivery service usage, having food delivery service apps downloaded to their smartphone suggests usage of these apps. Our study adds to the small existing body of evidence to suggest that food delivery service usage may be higher among people identifying as racial/ethnic minority groups.

## Implications of digital media marketing and food company app use

While most of the sampled youth indicated they had not liked, shared, or followed food companies on social media or did not have food company or food delivery service apps on their smartphones, approximately one-fifth to one-quarter did report interacting with companies on social media and downloading their apps. A study by our group found that that in 2020, 40 food brands were mentioned on Twitter, Reddit, Tumblr, and YouTube a combined 16.85 million times and reached approximately 42.24 billion users, with most posts being for fast food restaurants and sugar sweetened beverages [38]. Previous research has also demonstrated that youth's interactions with brands on social media mimics how they interact with their peers on social media, which is concerning because brands typically promote unhealthy products [39]. That study also found that youth preferred social media food advertisements with many likes, underscoring the powerful effect of social norms on youth's behaviours. Exposure to social media advertisements with more likes may

also increase youth's willingness to purchase the advertised products and reinforce brand loyalty [39]. The power of peer endorsement is also supported by a systematic review that found liking and sharing branded content on social media may have more influence on young people's attitudes and intended and actual consumption than digital media marketing owned or paid for by unhealthy food, alcohol or tobacco companies [40].

The large number of participants who reported having food company apps downloaded to their smartphones is also concerning given the volume and breadth of personal information that these companies collect on app users, including children. A recent study with children in Canada aged 9–12 years found that fast-food and dine-in restaurant apps are collecting a variety of data on child users, such as their personal information (e.g., name, birth date, country of residence, email address, preferred language), food preferences and purchasing habits (e.g., preferred foods, food order history, purchases made due to in-app promotions) [33]. In some cases, food company apps also collected information about the mobile operating system of the user's device, the internet provider, their frequency of app visits and clicks within the app, and the frequency and types of notifications from the app received and viewed by the user [33]. Adding to the concern is the fact that mobile apps, including those that are child-directed, often share users' data with third parties [41]. Previous research has also demonstrated that youth are exposed to various food marketing strategies on restaurant and food delivery service apps, with price discounts, combo deals and strategic placement of selected food products within the app being particularly common [42,43].

Overall, the findings of our study in combination with those of previous research reinforce the need for government policies concerning food marketing to extend to the digital environment, including to food company apps and food delivery service apps. Additional strategies such as parental involvement and bolstering youth's media literacy may also play a role. Specific examples may include teaching youth to think critically about food marketing and digital media messages, monitoring youth's digital media use, and encouraging open discussions about digital media and food marketing [44,45]. Such strategies reliant on individual behavioural changes are, however, not without challenges, such as the additional burden placed on parents and educators, the lack of a standardized marketing literacy curriculum in schools, and constraints related to finances, time or other resources [45]. Population-level policies such as regulations on digital marketing to youth – including the marketing of unhealthy foods – have the potential to ensure that all individuals benefit equally, irrespective of their socioeconomic status, geographic location or other social determinants of health [46]. Qualitative research to better understand why certain sociodemographic groups are more likely to engage with digital food marketing and consumer input into policy development are needed to inform such policies.

## Strengths and limitations

This study provided a novel examination of Canadian youth's engagement with food companies via digital media and explored differences between sociodemographic groups. This work was strengthened by the inclusion of gender, age group, race/ethnicity and income adequacy in the models, given that previous work has indicated these characteristics may influence youth's engagement with food companies on social media [9,18]. There are, however, important limitations to this study. Firstly, because this research relied on self-reported data in a sample of youth, there was potential for measurement error. Furthermore, the sampling strategy employed in this study aimed to ensure approximately equal numbers of respondents in each age group (10–12 years, 13–17 years), sex (female, male) and province (Ontario, Quebec) and therefore cannot be considered representative, limiting generalizability. Another limitation is the fact that this was a cross-sectional study conducted at a single point in time, which may have contributed to residual confounding. Moreover, the cross-sectional design prevented causal interpretations; experimental designs may better assess directionality, as well as the potential health impacts of youth engaging with food companies through digital media. The survey also did not ask youth about the frequency, purpose or context of use of food company or food delivery service apps, which is an important limitation given these behaviours are likely to differ across sociodemographic groups. In addition, while the survey asked about social media usage and food marketing exposure on these platforms, it did not ask if participants had social

media accounts. Lastly, survey panels such as Leger commonly include a large proportion of White participants of higher socioeconomic status [47], as was the case in this study, where 75% of participants identified as White and 28% were from households with high perceived income adequacy, thereby limiting the generalizability of our findings and potentially biasing our results. These figures are, however, similar to national population statistics, with approximately 74% of Canadians identifying as White in the 2021 Canadian census and 23% reporting annual incomes of $100,000 or more as of 2022 [48,49].

## Conclusions

This study examined youth's engagement with food companies via digital media, including liking, sharing, or following these companies on social media and downloading the apps of food and food delivery service companies to their smartphones. The results indicate that about one-firth or one-quarter of the sampled youth liked, shared, or followed food companies on social media and/or had their apps on their smartphones, making them vulnerable to unhealthy food marketing and peer influence. Moreover, differences were observed between sociodemographic groups, with adolescents and racial/ethnic minority group found most likely to engage with food companies via digital media. These findings reinforce the need for food marketing policies to extend to the digital environment and specifically to food company apps and food delivery service apps. Such policies may include mandatory government-led regulations that restrict targeting of both children and adolescents through social media posts by food companies, and requiring stricter age verification processes on food company and food delivery service apps to help limit youth's exposure to food marketing via digital media. Monitoring of the impacts of such policies on the digital engagement of youth, including specific population subgroups such as adolescents and individuals from racial/ethnic minority groups, will also be critical.

## Supporting information

**S1 Data. Data table.**
(XLSX)

**S1 Table. The number and percentage of participants in the analytic sample who identified as each racial/ethnic group and as having each level of income adequacy (n = 1162).**
(PDF)

## Author contributions

**Conceptualization:** Laura Vergeer, Meghan Pritchard, Monique Potvin Kent.

**Formal analysis:** Laura Vergeer.

**Funding acquisition:** Elise Pauzé, Ashley Amson, Dana Lee Olstad, Monique Potvin Kent.

**Methodology:** Laura Vergeer, Carolina Soto, Elise Pauzé, Ashley Amson, Monique Potvin Kent.

**Supervision:** Monique Potvin Kent.

**Writing – original draft:** Laura Vergeer, Meghan Pritchard.

**Writing – review & editing:** Laura Vergeer, Meghan Pritchard, Carolina Soto, Elise Pauzé, Ashley Amson, Dana Lee Olstad, Monique Potvin Kent.

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
