## [Decision Letter · Decision Letter 0]

13 Aug 2025

Response to Reviewers
Revised Manuscript with Track Changes
Manuscript
**Journal Requirements:**

1. Please send a completed 'Competing Interests' statement, including any COIs declared by your co-authors. If you have no competing interests to declare, please state "The authors have declared that no competing interests exist". Otherwise please declare all competing interests beginning with the statement "I have read the journal's policy and the authors of this manuscript have the following competing interests:"

2. Please ensure that your Ethics Statement is available in its entirety at the beginning of your Methods section, under a subheading 'Ethics Statement'. It must include:

1) The name(s) of the Institutional Review Board(s) or Ethics Committee(s)

2) The approval number(s), or a statement that approval was granted by the named board(s) 

3) (for human participants/donors) - A statement that formal consent was obtained (must state whether verbal/written) OR the reason consent was not obtained (e.g. anonymity). 

NOTE: If child participants, the statement must declare that formal consent was obtained from the parent/guardian.

3. Please amend your detailed Financial Disclosure statement. This is published with the article. It must therefore be completed in full sentences and contain the exact wording you wish to be published.

i. State the initials, alongside each funding source, of each author to receive each grant.

4. In the online submission form, you indicated that “Data generated or analyzed during this study are available from the corresponding author upon reasonable request.”. 

3. Uploaded as supplementary information.

**Additional Editor Comments (if provided):**
**Reviewers' Comments:**

**Comments to the Author**

1. Does this manuscript meet PLOS Digital Health’s publication criteria?

Reviewer #1: Yes

Reviewer #2: No

Reviewer #3: Partly

2. Has the statistical analysis been performed appropriately and rigorously?

Reviewer #1: Yes

Reviewer #2: Yes

Reviewer #3: Yes

3. Have the authors made all data underlying the findings in their manuscript fully available (please refer to the Data Availability Statement at the start of the manuscript PDF file)?

Reviewer #1: Yes

Reviewer #2: No

Reviewer #3: Yes

4. Is the manuscript presented in an intelligible fashion and written in standard English?

Reviewer #1: Yes

Reviewer #2: Yes

Reviewer #3: Yes

Reviewer #1: The manuscript offers a valuable and timely contribution by examining Canadian youths’ engagement with food companies via digital media, a topic that is increasingly relevant in the context of rising childhood obesity and unregulated digital food marketing. The study is commendable for its large, stratified sample, well-defined variables, and clear statistical approach, and it importantly highlights sociodemographic disparities in digital marketing exposure. However, several improvements could enhance the clarity and impact of the work. First, the operational definition of "engagement" (i.e., liking/sharing/following food companies and having apps on smartphones) should be emphasized more explicitly throughout the abstract, methods, and results to distinguish passive exposure from active interaction. Second, while the study reports the presence of food-related apps, no data were collected on the frequency, purpose, or context of use—an important limitation that should be discussed in more depth, along with how these behaviors may differ across age or income groups. Additionally, the representativeness of the sample requires more nuanced discussion; given the overrepresentation of White and higher-income participants, potential biases in the findings should be critically evaluated. The manuscript also reports no significant gender differences in overall engagement, but interesting differences appear in brand preferences and motivations (e.g., girls were more likely to follow food companies because the food looks tasty). These gendered behavioral nuances warrant deeper exploration, possibly supported by qualitative insights or future research. Moreover, the cross-sectional design limits causal interpretations; the authors could strengthen the discussion by acknowledging how longitudinal or experimental designs might better assess directionality and health impacts of engagement. From a policy perspective, the manuscript does an excellent job advocating for tighter digital media regulations, but it could further elaborate on how schools, parents, and media literacy programs might also serve as protective factors against persuasive marketing. Finally, while data are available on request, the authors are encouraged to consider depositing de-identified datasets in an open-access repository to align with best practices in open science. Overall, with minor revisions focused on improving clarity, contextual depth, and discussion of limitations, this manuscript will make a strong contribution to the literature on digital food marketing and public health.

Reviewer #2: Thank you for the opportunity to review this manuscript.

In this work, the authors describe the results of an online survey that was administered to children and adolescents in Canada. The goal of the survey was to assess the digital exposure of Canadian children and adolescents to food marketing, via apps and social media.

The study tackles an interesting issue in public health. Overall, the data has been collected rigorously, and the statistical analysis has been performed adequately.

However, I have three significant concerns about the manuscript that, in my opinion, raise serious questions about its suitability for publication.

First, the authors drive some strong conclusions from their analyses which are not fully supported by the results. In particular, it is unclear from the analysis if the type of engagement with food marketing apps or social media recorded from survey participants may lead to unhealthy or risky behaviors. The manuscript does not report whether children or adolescents exposed to food marketing were more likely to eat unhealthy food or not. Therefore, while it might be interesting to characterize the engagement with food marketing, it is hard to derive conclusions, especially about policy. The evidence to support the need for policy interventions focused on food company apps and food delivery apps is not sufficiently clear from the data.

Second, it appears that the authors have already published the results related to my first concern. Upon review, I found that the survey data used in this study have been analyzed previously—on at least two occasions—with findings published in other journals (Refs. 25 and 26). Notably, Ref. 26 presents the very relationship that is absent from the current manuscript and that would be essential to strengthen its conclusions. The repeated use of the same dataset across (at least) three different manuscripts raises concerns about the originality of each contribution and, in turn, diminishes the scientific value of the current work. Furthermore, Ref. 16, authored by the same researchers, is still under review and not publicly accessible. However, its title suggests that it may closely overlap with the current manuscript in terms of content. Is Ref. 16 also based on the same survey data? If so, this raises a serious concern.

Finally, PLOS Digital Health requires all papers to adhere to the highest standards of open science. Unfortunately, the data used in this study is not available. Given that the data can be readily pseudo-anonymized and deposited in a public repository, this omission is difficult to justify. Open data is particularly crucial in this case, as it appears that multiple publications have already been derived from the same dataset. Without access to the underlying data, the results of this study—as well as those of the related publications—cannot be independently verified or reproduced.

Reviewer #3: Overall Assessment

The manuscript addresses an important public health concern by examining youth exposure to digital marketing in Canada. However, several methodological and interpretative limitations reduce the strength and generalizability of the findings. Below, I provide major and minor recommendations to improve the rigor, representativeness, and methodological alignment with current computing trends.

Major Comments

1. Sample Representativeness and Geographic Scope

o The dataset consists of 1,162 observations collected only from Ontario and Quebec. While these provinces are significant in population size, limiting the sample to two provinces raises concerns about geographic and demographic representativeness for the entire Canada, as shown in the title.

o Recommendation: Expand data collection to include additional areas to ensure the findings reflect diverse socio-economic, cultural, and regional contexts.

2. Sample Size and Statistical Power

o The relatively small sample size (n = 1,162) may limit statistical power and the ability to detect meaningful associations. It also underutilizes the potential of modern computational resources, which can be acquired from larger datasets.

o Recommendation: Increase the sample size in future studies to enhance statistical reliability.

3. Policy Recommendations and Rationale

o The author’s primary policy recommendation is to restrict youth exposure to digital marketing. While this is a potentially effective intervention, the manuscript does not sufficiently justify why this measure is prioritized over, or in place of, other possible strategies.

o Recommendation: Provide a balanced discussion of alternative or complementary policy measures, such as promoting healthier product reformulation, implementing targeted educational campaigns, or incentivizing corporate responsibility initiatives. This would strengthen the policy relevance and applicability of the findings.

4. Methodological Approach and Model Selection

o The study employs traditional logistic regression, which is appropriate for binary classification tasks. However, relying solely on this approach may limit the model’s ability to capture complex, non-linear relationships in the data.

o Recommendation: Consider incorporating state-of-the-art machine learning models, such as TabNet with explainability features, in parallel with logistic regression. This would (a) improve predictive accuracy, (b) provide richer insights into feature importance, and (c) align the analytical approach with current trends in computational modeling.

**Do you want your identity to be public for this peer review?** For information about this choice, including consent withdrawal, please see our Privacy Policy

Reviewer #1: **Yes: ** Tooba Adil

Reviewer #2: No

Reviewer #3: No

**Figure resubmission:****Reproducibility:** To enhance the reproducibility of your results, we recommend that authors of applicable studies deposit laboratory protocols in protocols.io, where a protocol can be assigned its own identifier (DOI) such that it can be cited independently in the future. Additionally, PLOS ONE offers an option to publish peer-reviewed clinical study protocols. Read more information on sharing protocols at https://plos.org/protocols?utm_medium=editorial-email&utm_source=authorletters&utm_campaign=protocols

---

## [Decision Letter · Decision Letter 1]

3 Nov 2025

Response to Reviewers
Revised Manuscript with Track Changes
Manuscript
**Journal Requirements:**
**Reviewers' Comments:**

**Comments to the Author**

Reviewer #4: (No Response)

Reviewer #5: All comments have been addressed

Reviewer #6: (No Response)

publication criteria?

Reviewer #4: Yes

Reviewer #5: Yes

Reviewer #6: Yes

3. Has the statistical analysis been performed appropriately and rigorously?

Reviewer #4: Yes

Reviewer #5: Yes

Reviewer #6: Yes

4. Have the authors made all data underlying the findings in their manuscript fully available (please refer to the Data Availability Statement at the start of the manuscript PDF file)?

Reviewer #4: Yes

Reviewer #5: Yes

Reviewer #6: Yes

5. Is the manuscript presented in an intelligible fashion and written in standard English?

Reviewer #4: Yes

Reviewer #5: Yes

Reviewer #6: Yes

Reviewer #4: This study shows a sound methodology, and the contribution is important. To note the comments and responses from the previous reviewers:

1. A lack of an outcome of unhealthy eating. The discussion of studies that note a relationship between marketing exposure and unhealthy eating support the outcome, but it does reduce the inferences that this study can make.

2. Regarding previous publications on this dataset. The authors distinguish between previous research about “exposure” to digital marketing and the current study exploring “engagement” with digital marketing. The current analysis is limited engagement specifically.

3. The choice to undertake logistic regression only limits the analysis’ ability to explore the factors implementing the outcome. But I accept the lack of feasibility to address this in the current study and note the points addressed in the discussion.

The authors have responded to reviewers’ comments adequately and provide discussion of the limitations. The accepted limitations restrict the contribution of this research to a board association between some sociodemographic factors and digital food marketing engagement, with difficulty to extrapolate much further. However, there is a need for contributions in this area and the research feeds into a desire for policy intervention. The translational potential suggests this contribution is important and worthwhile, despite the limitations.

An additional note is that more focus should be applied to future qualitative research as a discussion point. If a key outcome of this research is to result in policy development and implementation, a better understand of “why” certain groups are more likely to engage in digital food marketing, and consumer input into policy development, is required for effective policy. Qualitative research is a good option, and recommendations should involve the inclusion of consumers in future policy development.

Reviewer #5: The authors have addressed all prior points thoroughly. I have no further comments at this stage. Congratulations on the careful and well-executed manuscript.

Reviewer #6: Excellent work. Please see minor revisions attached.

**Do you want your identity to be public for this peer review?** For information about this choice, including consent withdrawal, please see our Privacy Policy

Reviewer #4: **Yes: ** Calum Nicholson

Reviewer #5: **Yes: ** Shruti Muralidharan

Reviewer #6: No

**Figure resubmission:**

**Reproducibility:**To enhance the reproducibility of your results, we recommend that authors of applicable studies deposit laboratory protocols in protocols.io, where a protocol can be assigned its own identifier (DOI) such that it can be cited independently in the future. Additionally, PLOS ONE offers an option to publish peer-reviewed clinical study protocols. Read more information on sharing protocols at https://plos.org/protocols?utm_medium=editorial-email&utm_source=authorletters&utm_campaign=protocols

---

## [Editor Report · Decision Letter 2]

11 Dec 2025

Examining Canadian youth’s engagement with food companies via digital media

PDIG-D-25-00443R2

Dear Dr Potvin Kent,

We are pleased to inform you that your manuscript 'Examining Canadian youth’s engagement with food companies via digital media' has been provisionally accepted for publication in PLOS Digital Health.

Best regards,

Onicio Batista Leal-Neto

Academic Editor

PLOS Digital Health